# Thermal desorption as a high removal remediation technique for soils contaminated with per- and polyfluoroalkyl substances (PFASs)

**M. Sörengård**[ID]*, **A-S. Lindh, L. Ahrens**

Department of Aquatic Sciences and Assessment, Swedish University of Agricultural Sciences (SLU), Uppsala, Sweden

* mattias.sorengard@slu.se

**Data Availability Statement:** All relevant data are within the paper and its Supporting Information files.

## Abstract

Soils contaminated with per- and polyfluoroalkyl substances (PFASs) are an important source for impacting drinking water delivery systems and surface water bodies world-wide, posing an urgent risk to human health and environmental quality. However, few treatment techniques have been tested for PFAS-contaminated soil hotspots. This study investigated the possibility of thermal desorption as a possible technique to remediate soils contaminated with multiple PFASs. Two fortified soils ($\sum_9$PFAS $\approx$ 4 mg kg$^{-1}$) and one field-contaminated soil ($\sum_9$PFAS $\approx$ 0.025 mg kg$^{-1}$) were subjected to a 75-min thermal treatment at temperatures ranging from 150 to 550°C. Soil concentrations of PFASs showed a significant decrease at 350°C, with the $\sum_9$PFAS concentration decreasing by, on average, 43% and 79% in the fortified and field contaminated soils, respectively. At 450°C, >99% of PFASs were removed from the fortified soils, while at 550°C the fraction removed ranged between 71 and 99% for the field contaminated soil. In the field contaminated soil, PFAS classes with functional groups of sulfonates (PFSAs) and sulfonamides (FOSAs) showed higher removal than the perfluoroalkyl carboxylates (PFCAs). Thus thermal desorption has the potential to remove a wide variety of PFASs from soil, although more studies are needed to investigate the cost-effectiveness, creation of transformation products, and air-phase vacuum filtration techniques.

## 1. Introduction

Per- and polyfluoroalkyl substances (PFASs) are a large group of partially or completely fluorinated organic compounds that vary in structure and chemical properties and are generally persistent to thermal, chemical, and biological degradation [1, 2]. Concerns have been raised because of their ubiquitous distribution in the environment, high persistency, bioaccumulation potential, and adverse effects on humans and biota [3–8]. A common point source is the unregulated usage of PFAS-containing aqueous film-forming foams (AFFFs) at firefighter

**Funding:** M.S. VINNOVA 2015–03561 https://
www.vinnova.se/en/ The funders had no role in
study design, data collection and analysis, decision
to publish, or preparation of the manuscript.

**Competing interests:** The authors have declared
that no competing interests exist.

training facilities [9–11]. Although the use of PFAS-containing AFFFs is now restricted, the PFASs still present in contaminated soil are unsolicitedly leaching to the environment [12–14] and potentially polluting drinking water sources, e.g., in Japan [15], Germany [16], and Sweden [17]. Thus, there is an urgent need to remediate PFAS-contaminated hotspot areas and although a few soil guideline values have been set for PFASs, for example in Australia and New Zealand (i.e. 0.009, 2 and 20 mg kg$^{-1}$, depending on soil usage [18]). However, several countries have set drinking water guideline values for PFASs, for example in USA (i.e. 70 ng L$^{-1}$ for PFOS and PFOA) and Sweden 90 ng L$^{-1}$ for $\Sigma_{11}$PFASs), which are forcing problem owners to remediate their PFAS-contaminated soil [19].

The extreme challenges of PFAS-contaminated soil remediation has been acknowledged [20, 21] and multiple technologies have been reviewed. Suggested remediation methods for PFAS-contaminated soil are stabilization technologies [22–26], electrodialytical remediation [27] and phytoremediation [28]. However, soil stabilization methods do not provide a long-term solution and phytoremediation is a slow and long-term approach [20]. In a recent review article, Mahinroosta and Senevirathna [21] showed that there is a lack of laboratory-scale and field-scale studies of soil remediation for PFASs.

A conventional treatment method for soil is incineration, a costly but efficient *ex situ* treatment regarding high removal in which PFASs are destroyed by combusting the contaminated soil [29]. Fluorotelomer-based acrylic polymer waste and PFAS-contaminated sewage sludge have been reported to degrade PFASs successfully at 725˚C [30–33], although others have found that complete degradation of PFASs requires temperatures of 900–1100˚C [33–35]. Another viable thermal treatment method for contaminated solids is thermal desorption [36], where the solid is heated *ex situ* or *in situ* [29] and the vaporized contaminants partition to the air phase, from which they can be removed by air filters [37]. The technique is considered to be less energy-demanding than incineration, can achieve high removal [29], and is generally applicable for organic contaminants [38]. Thermal desorption has previously been shown to successfully remove persistent soil organic pollutants such as polyaromatic hydrocarbons (PAHs) and polychlorinated biphenyls (PCBs) at 500˚C [39], and PFAS thermal desorption from the soil phase has been observed at 350˚C after 10 days[40].

The aim of this study was therefore to further evaluate, whether thermal desorption is a viable remediation method for removal of PFASs in contaminated soil and to identify critical variables such as optimal temperature, soil texture, treatment times and fortified vs. natural aged contaminated soil. Specific objectives were to: i) evaluate whether thermal desorption can be used to remove PFASs from contaminated soil, ii) determine the removal for 9 commonly found and regulated PFASs,and iii) identify the temperatures required for thermal desorption of PFASs from different types of soils with different PFAS contamination levels.

## 2. Material and methods

### 2.1 Target compounds

The target PFASs comprised: six perfluoroalkyl carboxylates (PFCAs), namely perfluorobutanoate (PFBA), perfluorohexanoate (PFHxA), perfluorooctanoate (PFOA), perfluorononanoate (PFNA), perfluorodecanoate (PFDA), and perfluoroundecanoate (PFUnDA); two perfluoroalkyl sulfonates (PFSAs), namely perfluorohexane sulfonates (PFHxS) and perfluorooctane sulfonates (PFOS); and one perfluorooctanesulfonamide (FOSA) (purity >99%, Wellington Laboratories, Guelph, ON). A total of nine isotopically labeled internal standards (IS) were used: $^{13}C_4$-PFBA, $^{13}C_2$-PFHxA, $^{13}C_4$-PFOA, $^{13}C_5$-PFNA, $^{13}C_2$-PFDA, $^{13}C_2$-PFUnDA, $^{18}O_2$-PFHxS, $^{13}C_4$-PFOS, and $^{13}C_8$-FOSA (purity >99%, Wellington Laboratories, Guelph, ON).

## 2.2 Experimental design

The thermal desorption technique was assessed on bench-scale using two Swedish soils, a loamy sand soil from Högåsa and a clay soil from Vreta Kloster, both sampled at 0.35–0.45 m depth (for soil characterization, see [41] and laboratory-fortified with PFASs. In addition, a silty clay soil at a fire-fighter training facility known to be contaminated with PFAS-containing AFFFs, located at Stockholm Arlanda Airport, Sweden [28], was sampled at 0.10–0.30 m depth. The PFAS-fortified soils were separately fortified with a mixture of 11 PFASs, which resulted in a concentration of 600 $\mu$g kg$^{-1}$ for individual PFASs, and then aged for two months before the start of the experiment. The aging was performed by shaking (end-over-end, 100 rpm) a slurry of 0.5 kg fortified soil and 1 L of Millipore water for two weeks, freeze-drying the slurry for one week, and then shaking (end-over-end, 100 rpm) the dry soil for two weeks and storing it at 25˚C until use in the experiment.

The soils were separately freeze-dried (7 days) and homogenized with a mortar, and 4 g per sample were placed in amber glass bottles (40 mL, diameter 95 mm x height 27.5 mm; GENE-TEC, Sweden). Each soil was treated in experimental triplicates at 150˚C, 250˚C, 350˚C, 450˚C, or 550˚C ($n$ = 3 in each case) for 15, 45 and 75 min, respectively, using a high-temperature furnace (ThermoLyne$^{TM}$ 62700 Furnace, 19 cm x 22 cm x 33 cm). Previous studies have used treatment times ranging between 20 and 60 min for PCBs and PAHs [42–44]. Negative controls (soil samples treated, but not fortified with PFASs) ($n$ = 3) and positive controls (soil samples fortified with PFASs, but not treated) were included as reference and quality controls for each contaminated soil ($n$ = 9). All samples were stored air-tight at 4˚C before further analysis.

## 2.3 Sample preparation, analysis, and quality control

The soil samples were analyzed for PFASs according to a validated method, as described elsewhere [45]. In brief, liquid-solid extraction was used with 3.0 ± 0.2 g of freeze-dried (7 days) solid sample and 30 mL of methanol fortified with 100 $\mu$L of an IS mixture (c = 0.010 $\mu$g mL$^{-1}$). The eluent was concentrated by a nitrogen gas stream to 500 $\mu$L and the aliquot was fortified with 500 $\mu$L Millipore water (Millipore, Germany) and transferred to an Eppendorf tube (Eppendorf, Germany) for clean-up using 25 mg ENVICarb 120/400 (Supelco, USA) and glacial acetic acid (Merck, Germany). The tubes were then vortexed and centrifuged at 15000 rpm for 15 min. All samples were filtered with recycled cellulose syringe filters (Sartorius, 0.45 $\mu$m) into 1.5 mL auto-injector brown glass vials (Eppendorf, Germany).

Ultra-high performance liquid chromatography-tandem mass spectroscopy (UHPLC-MS/MS) (Quantiva TSQ; Thermo Fisher, MA, USA) was used for the chemical analysis. The injection volume was 10 $\mu$L separated on a BEH-C18 column (1.7 $\mu$m, 50 mm, Waters), with a run time of 12 min using methanol and Millipore water with 5 mM ammonium acetate (purity >99.99%, Sigma-Aldrich) as eluents. An eight-point calibration curve ranging between 0.01 and 100 ng mL$^{-1}$ with a linear fit ($R^2$) >0.99 was used for quantification. The data were evaluated using TraceFinder$^{TM}$ software (Thermo Fisher, MA, USA). Since no PFASs were present in the laboratory blanks ($n$ = 3), the limit of detection (LOD) was set to the lowest quantifiable calibration point with a signal to noise ratio >3. No PFASs were detected in laboratory blanks above LOD, and therefore, method detection limits (MDL) were set to LOD. The MDLs ranged between 0.003 (PFHxA and PFNA) and 0.2 (PFOS) $\mu$g kg$^{-1}$ dry weight (dw). The internal standard dilution method was used to compensate for losses or matrix effects. The relative recoveries were calculated as the measured concentration in fortified clay and sand reference samples compared with the theoretical fortification concentration (600 $\mu$g kg dw). The values obtained ranged between 48% (PFUnDA) and 130% (FOSA) (average 78%). The average

relative standard deviation (all samples were performed in experimental triplicates ($n$ = 3)) for the individual PFASs was 15 ± 8% and 11 ± 4.5% in the fieldfield contaminated soil and fortified soils, respectively. As a positive control, a non-fortified sandy soil was treated together with the contaminated samples at all temperatures ($n$ = 3). It was found that only PFBA displayed a concentration >0.1% of the concentration in the treated fortified soils, foremost at the lower treatment temperatures (5.0 ± 0.92% and 0.16 ± 0.016% difference compared with the PFBA concentrations at 150°C and 550°C, respectively) (Table S1 in S1 Appendix.

## 3 Results and discussion

The fraction removed PFASs from the soil generally increased with the treatment temperature, but was dependent on the soil type, soil initial concentration, and PFAS characteristics (Fig 1). Comparing the fortified clay and loamy sand soils, at 350°C PFCAs and FOSA were removed to >99%, while the PFSAs showed removal ranging between 51% (PFHxS) and 66% (PFOS). At 450°C, the removal was >99% for all PFASs in both PFAS-fortified soils. This indicates that the functional group is an important parameter influencing the desorption potential of PFASs. However, the difference between PFSAs and PFCAs cannot be explained by the vapor pressure, since the vapor pressure constants (log $P_L$ [log Pa]) were similar for the two groups (range 0.83–2.9 for the $C_4$-$C_8$ PFSA and 0.82–3.1 for the $C_3$-$C_{10}$ PFCAs) [46]. However, these vapor pressure constants were estimated using the COSMOtherm model for chemical properties and can be biased due to the analysis being limited to non-ionized neutral forms of the PFASs. The PFCAs and PFSAs included in this study generally have an acid dissociation constant ($pK_a$) <2 [47], with the exception of FOSA, which has $pK_a$ = 6.2–6.5 [48], meaning that they are predominantly protonated anions at the pHs in the tested soils. On the other hand, previous studies have shown that PFSAs are more strongly sorbed than PFCAs to soil and sediments [49–51], which might result in lower desorption potential for PFSAs. However, PFOS and PFOA are reported to show similar desorption behavior in soil [52]. In contrast to the two fortified soils, the fraction removed PFASs from the field contaminated soil were lower at all treatment temperatures tested. At 350°C, $\Sigma_9$PFASs removal was 43%, compared with 71% and 87% for the fortified clay and sand soil, respectively. At 450°C, 99% of PFSAs (PFHxS, PFOS) were removed from the field contaminated soil, similarly to the fortified soils. At 550°C, PFPeA was removed to >97%, but other PFCAs were only removed to 71–93%, from the field contaminated soil. The lower desorption potential of PFASs in the field contaminated soil compared with the fortified soils could be explained by the lower concentration of PFASs or/ and stronger sorption of PFASs in the naturally aged soil than in the fortified soils [53]. The shorter-chained PFCA (PFBA) even showed negative removal for the field contaminated soil at 150–450°C, which might be explained by the presence of unidentified precursor compounds that degraded into PFBA [54, 55]. This is a particular concern, as the use of shorter-chain PFASs has increased since the ban on $C_8$-based PFASs in AFFF [56]. Except of PFBA, no degradation products could be observed throughout the experiment, i.e. increase of shorter chain homologues as a product of longer chain PFASs, which has been observed in other PFAS degradation studies [57, 58]. This indicates that the main removal mechanism observed in this study is thermal desorption and not degradation.

When comparing the treatment times at 15, 45 and 75 min the desorption behavior was similar between the three soils (Tables S2-4 in SI). The PFAS concentrations in the soil decreased with increasing treatment time for almost all temperatures (except for 550°C), Fig 2. This experiment showed that the optimal temperature and treatment time for thermal desorption of PFASs is between 350°C and 450°C, and between 15 and 45 min, which is in agreement with a previous study using 350°C but 10 days treatment time [40]. In addition, a treatment

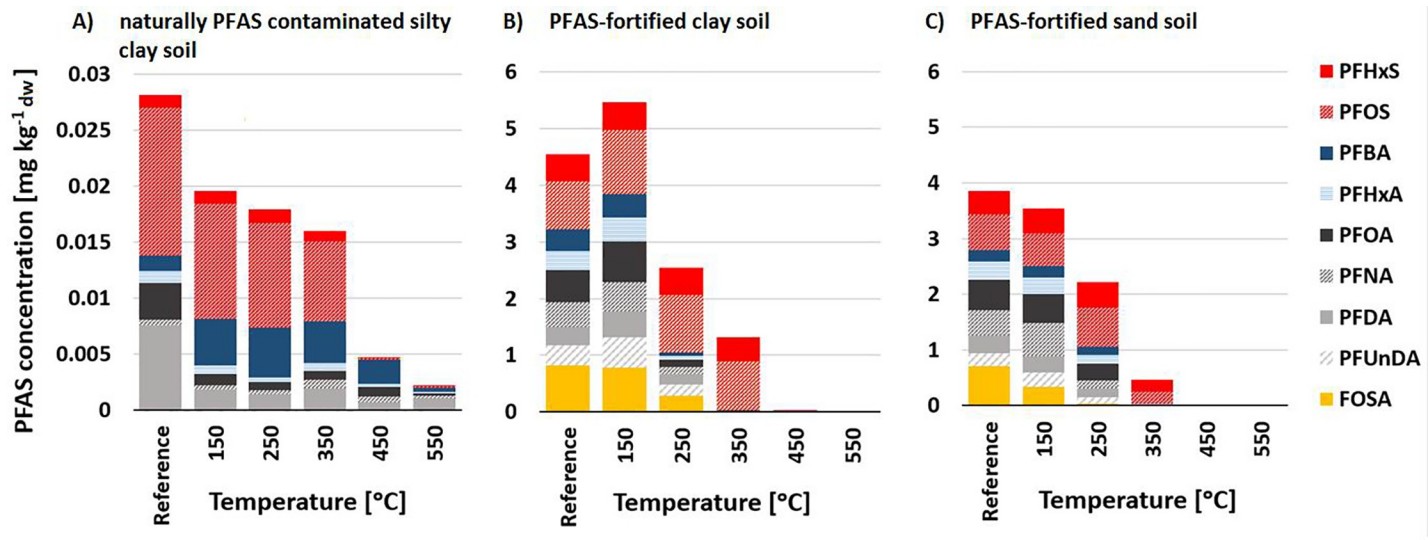

**Fig 1.** Individual PFAS concentrations after thermal desorption for 75 min at different treatment temperatures of A) naturally PFAS-contaminated soil from Stockholm Arlanda Airport, Sweden, B) fortified clay soil, and C) fortified loamy sand soil.

time of over 45 min for thermal desorption of PFASs is not necessary to minimize the energy demand of this treatment option.

Ultimately, the results indicate that soils polluted with mixtures of PFASs can be treated with high removal using thermal desorption at temperatures above 450˚C. This is a higher temperature than for other organic pollutants such as PCBs and PAHs, which showed high removal at 400˚C [43, 44]. The high removal using thermal desorption can be compared to stabilization techniques where the leaching of PFASs to the aqueous phase can be reduced by >99% [59, 26], but comparably higher than using granulated activated carbon or anion exchange for PFAS removal in drinking water [60]. Although soil stabilization techniques have shown high removal from leachate water, their long-term performance is still unknown, while thermal desorption removes PFASs from the soil and thereby reducing the risks of leaching into the aquatic environment. In addition, the treatment of contaminated soil using thermal

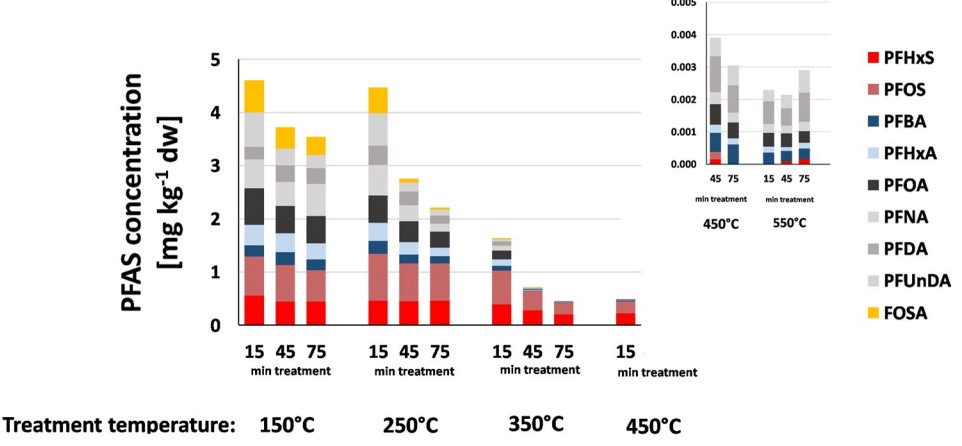

**Fig 2. Individual PFAS concentrations after thermal desorption for 15, 45 and 75 min at different treatment temperatures of fortified loamy sand soil.**

desorption has the advantage of also treating other co-contaminants often associated with PFAS-contaminated sites, e.g., non-aqueous phase liquid (NAPL), nonfluorinated AFFF surfactants, polyhalogenated compounds (PHCs), volatile organic compounds (VOCs), PCBs, PAHs, and metals [61, 43, 62], although heavy metals are not known to be affected by thermal desorption [63]. Further studies are required to test the thermal treatment methodology at field scale and assess possible degradation compounds, determine removal, and evaluate air-phase vacuum extraction and air filtration. Measurement of PFASs in the air phase is also needed, in order to enable mass balance calculations post-treatment to identify PFAS degradation or formation of PFAS degradation products. As a concluding remark, it should be noted that the treatment temperatures of 450–550˚C is not of insignificant magnitude and energy costs will ultimately be a limiting factor for problem owners to consider when comparing thermal desorption with other remediation methods for PFASs in soil or other solid materials.

## Supporting information

**S1 Appendix.**
(DOCX)

## Author Contributions

**Conceptualization:** M. Sörengård, L. Ahrens.

**Formal analysis:** A-S. Lindh.

**Investigation:** M. Sörengård, A-S. Lindh.

**Methodology:** M. Sörengård, L. Ahrens.

**Project administration:** L. Ahrens.

**Resources:** L. Ahrens.

**Supervision:** M. Sörengård, L. Ahrens.

**Validation:** L. Ahrens.

**Visualization:** M. Sörengård, L. Ahrens.

**Writing – original draft:** M. Sörengård.

**Writing – review & editing:** M. Sörengård, A-S. Lindh, L. Ahrens.

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
