## [Decision Letter · Decision Letter 0]

12 Mar 2020

PONE-D-19-33947

Thermal desorption as an effective remediation technique for soils contaminated with per- and polyfluoroalkyl substances (PFASs)

PLOS ONE

Dear Mr sörengård,

Thank you for submitting your manuscript to PLOS ONE. After careful consideration, we feel that it has merit but does not fully meet PLOS ONE’s publication criteria as it currently stands. Therefore, we invite you to submit a revised version of the manuscript that addresses the points raised during the review process.

We would appreciate receiving your revised manuscript by Apr 26 2020 11:59PM. To enhance the reproducibility of your results, we recommend that if applicable you deposit your laboratory protocols in protocols.io, where a protocol can be assigned its own identifier (DOI) such that it can be cited independently in the future. For instructions see: http://journals.plos.org/plosone/s/submission-guidelines#loc-laboratory-protocols

We look forward to receiving your revised manuscript.

Kind regards,

Jamie C. DeWitt

Academic Editor

PLOS ONE

Additional Editor Comments (if provided):

Please note that both reviewers ask for additional experimental design and results information to be provided in a revision.

Journal Requirements:

Please ensure that your manuscript meets PLOS ONE's style requirements, including those for file naming. The PLOS ONE style templates can be found at http://www.plosone.org/attachments/PLOSOne_formatting_sample_main_body.pdf and http://www.plosone.org/attachments/PLOSOne_formatting_sample_title_authors_affiliations.pdf

Reviewers' comments:

Reviewer's Responses to Questions

**Comments to the Author**

1. Is the manuscript technically sound, and do the data support the conclusions?

Reviewer #1: Partly

Reviewer #2: Yes

2. Has the statistical analysis been performed appropriately and rigorously? 

Reviewer #1: I Don't Know

Reviewer #2: No

3. Have the authors made all data underlying the findings in their manuscript fully available?

Reviewer #1: No

Reviewer #2: Yes

4. Is the manuscript presented in an intelligible fashion and written in standard English?

Reviewer #1: Yes

Reviewer #2: Yes

5. Review Comments to the Author

Reviewer #1: General

I would like to commend the authors of this manuscript for addressing the important issue of assessment of viable remediation techniques for PFAS-impacted soil. I view thermal desorption as a treatment technique that merits further investigation, so I think studies of this type are important to the field. Additionally, the manuscript is very clearly written, logically organized, and easy to follow. At points in the manuscript the authors acknowledge that this study is not a complete assessment of the feasibility of thermal desorption as a treatment technique, and I agree. However, the data strike me as being an extremely early and initial step in assessing thermal desorption, and results raise many questions. In some instances, publication of early results of this nature may be of value, but the need for publishing this data at this stage of the study is not immediately evident in the manuscript. So, my primary comment is that the authors should do more to demonstrate the role this work plays in a more complete assessment of thermal desorption of PFAS and why it is important to publish it now. More details regarding these recommendations as well as additional comments are provided in the line by line summary below.

Title/Abstract

• Lines 20-21 state that removal efficiencies were higher for PFSAs and sulfonamides, but the results section discusses the opposite. For example, lines 124-126 state that removal effieciecies of PFCAs and FOSA were >99% whereas removal efficiencies for PFHxS and PFOS were 51-66%.

Introduction

• The introduction is generally well-written, but given that the article is quite concise, I think there is room for the authors to add a short paragraph on needs for assessing the feasibility of thermal desorption as a treatment technique for PFAS-impacted soils and then outline how the current work fits into those needs.

• Lines 57-58 state that this is the first time anyone has evaluated thermal desorption as a remedation method for PFAS; however I know of at least one prior study: Crownover, E., Oberle, D., Kluger, M., & Heron, G. (2019). Perfluoroalkyl and polyfluoroalkyl substances thermal desorption evaluation. Remediation Journal, 29(4), 77-81.

• Lines 60-62 state that an objective of this study is to identify the temperatures required for thermal desorption of PFASs from soil; however, I am not sure that the design of the current study can achieve this objective. The authors have only confirmed a reduction of the concentrations of select PFAS in soil but have not confirmed their transfer to the vapor phase or monitored potential degradation products.

Methods

• Lines 90-93 discuss two types of control experiments; however, results from the positive controls are not presented.

• Lines 88-90 describe use of 20- and 60-min treatment times in studies of thermal desorption of other classes of compounds; however do these or other studies use similar temperatures? While I have seen other studies of in situ thermal desorption the elevated temperatures implemented in this study, it seems that use of lower temperatures are more the norm. This makes sense from a feasibility and cost standpoint. Thus, I think some comment on use of temperatures of this magnitude in prior lab studies and the feasibility of actually using this in the field is warranted. Particularly as the authors conclude that there is promise in the technique they are testing.

• Line 116-117 describe representative standard deviations, but do not indicate how these were determined. For example, was each sample analyzed using triplicate analysis?

• Along those lines, the methods section does not discuss use of experimental replicates.

Results and Discussion

• Line 153 states that the study results demonstrate that PFAS mixtures can be efficiently treated with thermal desorption. What defines efficient treatment? For example, are there regulatory criteria that final concentrations can be compared to? Is efficient an indicator of cost? Without a clearer indication of the criteria that the authors are using to define efficient, it is unclear if the study results support this conclusion.

• Lines 156-161 is not clear to me. By saying that thermal desorption has the advantage of “avoiding simultaneous treatment” of co-contaminants, it sounds like the authors are saying that co-contaminants would not be impacted by this treatment technique. Do the authors mean to say that it avoids the need for a separate treatment approach to address co-contaminants? In other words, that this technique would be capable of addressing PFAS+co-contaminant mixtures with a single approach? Because some of the contaminants listed (i.e. VOCs) are treated with thermal desorption techniques.

• Line 161 cites Guemiza et al. 2017, but I cannot find this reference in the references cited section.

References

• Please see my comment regarding lines 57-58 which includes a reference that may be useful to cite in this work.

• I could not locate the reference Gitipour et al. 2015. I suggest adding a DOI number as done for previous references. If not available, a web address would be useful.

Figures and Tables:

• I recommend adding data for the positive control to Figure 1.

• I also recommend adding a table of numerical results. For researchers who would like to compare results of this study, include them as part of a review, etc. estimating concentrations from the graphical data would be imprecise. Additionally, a table would provide an opportunity to include the standard deviations of each analysis. Lastly, data for controls should be made available.

Reviewer #2: In this manuscript (PONE-D-19-33947), the authors present findings of a small scale study to determine if thermal desorption has efficacy as a technique to remediate soils containing per- and polyfluoroalkyl substances (PFAS).

Comments, questions, and suggestions.

1. Abstract, general. Can the authors really write that thermal desorption “removed” PFAS from soil? Aren’t the PFAS being degraded (as is the phraseology in parts of the manuscript) rather than being removed? The authors may want to clarify/consider changing the use of this word.

2. Abstract, lines 10-11. Is it just PFAS-contaminated soils that are impacting drinking water delivery systems and surface water bodies or are contaminated soils just one source of water contamination? The authors may want to clarify this statement.

3. Introduction. Line 34. Are PFAS in AFFF restricted globally or in specific regions/countries? Some countries still allow PFAS in AFFF. The authors are asked to clarify this statement.

4. Materials and Methods, section 2.1. Could the authors please provide a rationale for the choice of the selection of PFAS?

5. Materials and Methods, general. The authors have appeared to have left out information on positive and negative control experiments as well as experimental replicates (were there any and if so, how many?). Please add this information to the Materials and Methods.

6. Materials and Methods, general. Did the authors perform any statistical analysis within soil types and among temperatures or among soil types within temperatures to determine if values differed statistically? If not, it is recommended that the authors provide those details.

7. Results and Discussion, general. Do any countries have regulatory standards or advisories for levels of PFAS in soils? If so, would the technique described in the manuscript reduce PFAS in soils to such regulatory standards or advisories? If there are not standards or advisories, what criteria do the authors use for efficiency? In other words, how do the authors know that thermal desorption is efficient? By what standard(s) are the authors comparing their results to?

8. Results and Discussion, lines 153-161. This sentence is unclear. If PCBs and PAHs have high removal “efficiencies” (again, clarification of “efficiency” is warranted) at 400-C, then why wouldn’t they be treated at a higher temperature? Wouldn’t an ideal treatment for contaminated soils treat multiple contaminants? This section seems counter-intuitive.

6. PLOS authors have the option to publish the peer review history of their article (what does this mean?). If published, this will include your full peer review and any attached files.

Reviewer #1: No

Reviewer #2: No

---

## [Author Response · Author response to Decision Letter 0]

19 Mar 2020

see attached file

or:

Response to Reviewers

PONE-D-19-33947

Thermal desorption as an effective remediation technique for soils contaminated with per- and polyfluoroalkyl substances (PFASs)

PLOS ONE

Title changed to:

Thermal desorption as a high removal remediation technique for soil contaminated with per- and polyfluoroalkyl substances (PFASs)

We thank reviewers and editor for considering this paper in PLOS ONE. Replies to reviewer’s comments (black) can be found in bolded blue color and changes in the manuscript is written in italic and referred with line numbers (in the tracked changes manuscript).

PLOS ONE

Reviewer #1: General

I would like to commend the authors of this manuscript for addressing the important issue of assessment of viable remediation techniques for PFAS-impacted soil. I view thermal desorption as a treatment technique that merits further investigation, so I think studies of this type are important to the field. Additionally, the manuscript is very clearly written, logically organized, and easy to follow. At points in the manuscript the authors acknowledge that this study is not a complete assessment of the feasibility of thermal desorption as a treatment technique, and I agree. However, the data strike me as being an extremely early and initial step in assessing thermal desorption, and results raise many questions. In some instances, publication of early results of this nature may be of value, but the need for publishing this data at this stage of the study is not immediately evident in the manuscript.

So, my primary comment is that the authors should do more to demonstrate the role this work plays in a more complete assessment of thermal desorption of PFAS and why it is important to publish it now. More details regarding these recommendations as well as additional comments are provided in the line by line summary below.

Title/Abstract

• Lines 20-21 state that removal efficiencies were higher for PFSAs and sulfonamides, but the results section discusses the opposite. For example, lines 124-126 state that removal efficiencies of PFCAs and FOSA were >99% whereas removal efficiencies for PFHxS and PFOS were 51-66%.

We thank you for the comment, and it is valid. The statement referred to naturally contaminated soil (Line 143-148), and not a general behavior. 

Changed to (Line 20-22)

“In the naturally contaminated soil, PFAS classes with functional groups of sulfonates (PFSAs) and sulfonamides (FOSAs) showed higher removal rate than the perfluoroalkyl carboxylates (PFCAs)”

Introduction

• The introduction is generally well-written, but given that the article is quite concise, I think there is room for the authors to add a short paragraph on needs for assessing the feasibility of thermal desorption as a treatment technique for PFAS-impacted soils and then outline how the current work fits into those needs.

In general, there is a big lack of experimental studies on PFAS soil remediation in comparison to PFAS water treatment studies.

We have now extended this section (Line 47-59):

“The extreme challenges of PFAS-contaminated soil remediation has been acknowledged (Ross et al., 2018; Mahinroosta and Senevirathna, 2020) and multiple technologies have been reviewed. Suggested remediation methods for PFAS-contaminated soil are stabilization technologies (Das et al., 2013; Hale et al., 2017; Kupryianchyk et al., 2016; McGregor, 2018; Sörengård et al., 2019), electrodialytical remediation (Sörengård et al., 2019b) and phytoremediation (Gobelius et al., 2017). However, soil stabilization methods do not provide a long-term solution and phytoremediation is a slow and long-term approach (Ross et al., 2018). In a recent review article, Mahinroosta and Senevirathna (2020) showed that there is a lack of laboratory-scale and field-scale studies of soil remediation for PFASs. “

• Lines 57-58 state that this is the first time anyone has evaluated thermal desorption as a remedation method for PFAS; however I know of at least one prior study: Crownover, E., Oberle, D., Kluger, M., & Heron, G. (2019). Perfluoroalkyl and polyfluoroalkyl substances thermal desorption evaluation. Remediation Journal, 29(4), 77-81.

We thank you for informing about this relatively new research study. This changes the novelty of this study (this study was executed in 2017, but we have been delayed in the publishing track), but we have reformulated the aims:

We have now extended this section (Line 78-81):

“The aim of this study was therefore to further evaluate, whether thermal desorption is a viable remediation method for removal of PFASs in contaminated soil and to identify critical variables such as optimal temperature, soil texture, treatment times and fortified vs. natural aged contaminated soil”

• Lines 60-62 state that an objective of this study is to identify the temperatures required for thermal desorption of PFASs from soil; however, I am not sure that the design of the current study can achieve this objective. The authors have only confirmed a reduction of the concentrations of select PFAS in soil but have not confirmed their transfer to the vapor phase or monitored potential degradation products.

Thank you for this comment. It is a valid point that we have not analyzed the air phase. This was because there are no (to our knowledge) established analytical method for high temperature air PFAS sampling, and this was outside the scope of this study. We have changed the wording in the aims (integrated in comment above). However, except for PFBA, no formation of shorter chain PFASs has been observed in this study, which suggest that thermal desorption and not degradation is occurring. Degradation of PFASs has been described at temperatures >900 °C (in this study a maximum of 550 °C was used). 

Following section has been added (Line 179-183):

“Except of PFBA, no degradation products could be observed throughout the experiment, i.e. increase of shorter chain homologues as a product of longer chain PFASs, which has been observed in other PFAS degradation studies (Y. Liu et al., 2015; Franke et al., 2019). This indicates that the main removal mechanism observed in this study is thermal desorption and not degradation.”

Methods

• Lines 90-93 discuss two types of control experiments; however, results from the positive controls are not presented.

The results of the control experiment are included in Table S1 in Supporting Information (SI)

• Lines 88-90 describe use of 20- and 60-min treatment times in studies of thermal desorption of other classes of compounds; however do these or other studies use similar temperatures?

“at 500°C “ is now specified

While I have seen other studies of in situ thermal desorption the elevated temperatures implemented in this study, it seems that use of lower temperatures are more the norm. This makes sense from a feasibility and cost standpoint. Thus, I think some comment on use of temperatures of this magnitude in prior lab studies and the feasibility of actually using this in the field is warranted. Particularly as the authors conclude that there is promise in the technique they are testing.

Feasibility and cost standpoint has been discussed by the authors. Accordingly, 500-550°C is a technical limit due to the high-energy costs. Eventually, a cost-benefit analysis has to be performed by the problem owners and regulators comparing thermal desorption with other treatment options.

Following section has been added (Line 220-224):

 “As a concluding remark, it should be noted that the treatment temperatures of 450-550°C is not of insignificant magnitude and energy costs will ultimately be a limiting factor for problem owners to consider when comparing thermal desorption with other remediation methods for PFASs in soil or other solid materials.”

• Line 116-117 describe representative standard deviations, but do not indicate how these were determined. For example, was each sample analyzed using triplicate analysis?

Thank you, triplicates was used and was denoted (n = 3), but has now been clarified in the text:

“The average relative standard deviation (all samples were performed in experimental triplicates (n = 3)) for the individual PFASs was 15 ± 8 % and 11 ± 4.5 % in the naturally contaminated soil and fortified soils, respectively.”

• Along those lines, the methods section does not discuss use of experimental replicates.

Thank you, triplicates was used and has now been clarified:

Each soil was treated in experimental triplicates at 150°C , 250°C , 350°C , 450°C, or 550°C (n = 3 in each case) for 75 min using a high-temperature furnace (ThermoLyneTM 62700 Furnace, 19 cm x 22 cm x 33 cm).

Results and Discussion

• Line 153 states that the study results demonstrate that PFAS mixtures can be efficiently treated with thermal desorption. What defines efficient treatment? 

Thank you for a valid comment, efficiency is referring to more than removal efficiency. Therefore

“with a high removal rate” has been added”

And “efficient” has been replaced with “high removal rates”

For example, are there regulatory criteria that final concentrations can be compared to? 

There are drinking water guidelines but very few guideline values for PFASs in soil. This has been now address in the introduction:

Following section has been added (Line 201-209):

“The high removal rates using thermal desorption can be compared to stabilization techniques, where the leaching of PFASs to the aqueous phase can be reduced by >99% (Kupryianchyk et al., 2016b; Sörengård et al., 2019a), but comparably higher than using granulated activated carbon or anion exchange filters for PFAS removal in drinking water (McCleaf et al., 2017). Although soil stabilization techniques have shown high removal rates, their long-term performance is still unknown, while thermal desorption removes PFASs from the soil and thereby reducing the risks of leaching into the aquatic environment”

Is efficient an indicator of cost? Without a clearer indication of the criteria that the authors are using to define efficient, it is unclear if the study results support this conclusion.

“Removal efficiency” has been replaced with “removal rate”

And “efficient” has been replaced with “high removal rates”

• Lines 156-161 is not clear to me. By saying that thermal desorption has the advantage of “avoiding simultaneous treatment” of co-contaminants, it sounds like the authors are saying that co-contaminants would not be impacted by this treatment technique. Do the authors mean to say that it avoids the need for a separate treatment approach to address co-contaminants? In other words, that this technique would be capable of addressing PFAS+co-contaminant mixtures with a single approach? Because some of the contaminants listed (i.e. VOCs) are treated with thermal desorption techniques.

Changed to (for clarification)(Line 209-215):

“In addition, the treatment of contaminated soil using thermal desorption has the advantage of also treating other co-contaminants often associated with PFAS-contaminated sites, e.g., non-aqueous phase liquid (NAPL), non fluorinated AFFF surfactants, polyhalogenated compounds (PHCs), volatile organic compounds (VOCs), PCBs, PAHs, and metals (Guelfo and Higgins, 2013; Qi et al., 2014; Yao et al., 2015), although heavy metals are not known to be affected by thermal desorption (Guemiza et al., 2017)”

• Line 161 cites Guemiza et al. 2017, but I cannot find this reference in the references cited section.

Changed

References

• Please see my comment regarding lines 57-58 which includes a reference that may be useful to cite in this work.

Changed

• I could not locate the reference Gitipour et al. 2015. I suggest adding a DOI number as done for previous references. If not available, a web address would be useful.

Removed

Figures and Tables:

• I recommend adding data for the positive control to Figure 1.

Now available in Table S1 in Supporting Snformation (SI)

• I also recommend adding a table of numerical results. For researchers who would like to compare results of this study, include them as part of a review, etc. estimating concentrations from the graphical data would be imprecise. Additionally, a table would provide an opportunity to include the standard deviations of each analysis. Lastly, data for controls should be made available.

Now available in Tables S2-4 in Supporting Information (SI)

Reviewer #2: In this manuscript (PONE-D-19-33947), the authors present findings of a small scale study to determine if thermal desorption has efficacy as a technique to remediate soils containing per- and polyfluoroalkyl substances (PFAS).

Comments, questions, and suggestions.

1. Abstract, general. Can the authors really write that thermal desorption “removed” PFAS from soil? Aren’t the PFAS being degraded (as is the phraseology in parts of the manuscript) rather than being removed? The authors may want to clarify/consider changing the use of this word.

We have changed the wording in the aim (integrated in comment above). However, except for PFBA, no formation of shorter chain PFASs has been observed in this study, which suggest that thermal desorption and not degradation is occurring. Degradation of PFASs has been described at temperatures >900 °C (in this study a maximum of 550 °C was used). 

No changes was made and we refer to following section (Line 62-66)

“Fluorotelomer-based acrylic polymer waste and PFAS-contaminated sewage sludge have been reported to degrade PFASs successfully at 725°C (Loganathan et al., 2007; Vecitis et al., 2009; Wang et al., 2013; Yamada et al., 2005), although others have found that complete degradation of PFASs requires temperatures of 900-1100°C (Yamada et al., 2005; Wang et al., 2015a; Watanabe et al., 2016).”

2. Abstract, lines 10-11. Is it just PFAS-contaminated soils that are impacting drinking water delivery systems and surface water bodies or are contaminated soils just one source of water contamination? The authors may want to clarify this statement.

The first sentence in the abstract has been modified to “Soils contaminated with per- and polyfluoroalkyl substances (PFASs) are an important source for impacting drinking water delivery systems…” 

No further changes was made and we refer to following section (Line 32-67) with no changes:

“A common point source is the unregulated usage of PFAS-containing aqueous film-forming foams (AFFFs) at firefighter training facilities (Anderson et al., 2016; Barzen-Hanson et al., 2017; Mejia-Avendaño et al., 2017). Although the use of PFAS-containing AFFFs is now restricted, the PFASs still present in contaminated soil are unsolicitedly leaching to the environment (Ahrens et al., 2015; Baduel et al., 2015; Filipovic et al., 2015) and potentially polluting drinking water sources, e.g., in Japan (Murakami et al., 2009), Germany (Gellrich et al., 2013), and Sweden (Li et al., 2018).”

3. Introduction. Line 34. Are PFAS in AFFF restricted globally or in specific regions/countries? Some countries still allow PFAS in AFFF. The authors are asked to clarify this statement.

Additional references of the global PFAS regulation in drinking water and soil has been added: (Gobelius et al., 2018; Heads of EPAs Australia and New Zealand (HEPA), 2018)

4. Materials and Methods, section 2.1. Could the authors please provide a rationale for the choice of the selection of PFAS?

Now specified in the aims (Line 83):

“determine the removal rates for 9 commonly found and regulated PFASs”

In addition, only a limited number of isotopically labeled PFAS are available as internal standards and we used perfectly matching standards for high quality quantification, see QA/QC. This generated low triplicate standard deviation and good relative recoveries.

5. Materials and Methods, general. The authors have appeared to have left out information on positive and negative control experiments as well as experimental replicates (were there any and if so, how many?). Please add this information to the Materials and Methods.

In formation on positive and negative control experiments as well as experimental replicates are now included in the Materials and Methods section:

Changed to (for clarification)(Line 110-112):

“Each soil was treated in experimental triplicates at 150°C , 250°C , 350°C , 450°C, or 550°C (n = 3 in each case) for 15, 45 and 75 min, respectively, using a high-temperature furnace (ThermoLyneTM 62700 Furnace, 19 cm x 22 cm x 33 cm).”

and (Line 141-143):

 “The average relative standard deviation (all samples were performed in experimental triplicates (n = 3)) for the individual PFASs was 15 ± 8 % and 11 ± 4.5 % in the naturally contaminated soil and fortified soils, respectively.”

6. Materials and Methods, general. Did the authors perform any statistical analysis within soil types and among temperatures or among soil types within temperatures to determine if values differed statistically? If not, it is recommended that the authors provide those details.

Triplicates were used for all samples and standard all deviations is now available in SI.

7. Results and Discussion, general. Do any countries have regulatory standards or advisories for levels of PFAS in soils? If so, would the technique described in the manuscript reduce PFAS in soils to such regulatory standards or advisories? If there are not standards or advisories, what criteria do the authors use for efficiency? In other words, how do the authors know that thermal desorption is efficient? By what standard(s) are the authors comparing their results to?

Adressed: see comments above reviewer #1

8. Results and Discussion, lines 153-161. This sentence is unclear. If PCBs and PAHs have high removal “efficiencies” (again, clarification of “efficiency” is warranted) at 400-C, then why wouldn’t they be treated at a higher temperature? Wouldn’t an ideal treatment for contaminated soils treat multiple contaminants? This section seems counter-intuitive.

This section has been revised. 

Figure 2 has been added:

Figure 2. Individual PFAS concentrations after thermal desorption for 15, 45 and 75 min at different treatment temperatures of fortified loamy sand soil.

And the following text has been added:

“When comparing the treatment times at 15, 45 and 75 min, the desorption behavior was similar between the three soils (Tables S2-4 in SI). The PFAS concentrations in the soil decreased with increasing treatment time for almost all temperatures (except for 550 °C) (Figure 2). This experiment showed that the optimal temperature and treatment time for thermal desorption of PFASs is between 350°C and 450°C, and between 15 and 45 min, which is in agreement with a previous study using 350°C but 10 days treatment time (Crownover et al., 2019). In addition, a treatment time of over 45 min for thermal desorption of PFASs is not necessary to minimize the energy demand of this treatment option.”

Additionally, Tables S1-4 have been included in the Supporting Information.

Ahrens, L., Norström, K., Viktor, T., Cousins, A.P., Josefsson, S., 2015. Stockholm Arlanda Airport as a source of per- and polyfluoroalkyl substances to water, sediment and fish. Chemosphere, Per- and Polyfluorinated Alkyl substances (PFASs) in materials, humans and the environment – current knowledge and scientific gaps. 129, 33–38. https://doi.org/10.1016/j.chemosphere.2014.03.136

Anderson, R.H., Long, G.C., Porter, R.C., Anderson, J.K., 2016. Occurrence of select perfluoroalkyl substances at U.S. Air Force aqueous film-forming foam release sites other than fire-training areas: Field-validation of critical fate and transport properties. Chemosphere 150, 678–685. https://doi.org/10.1016/j.chemosphere.2016.01.014

Baduel, C., Paxman, C.J., Mueller, J.F., 2015. Perfluoroalkyl substances in a firefighting training ground (FTG), distribution and potential future release. J. Hazard. Mater. 296, 46–53. https://doi.org/10.1016/j.jhazmat.2015.03.007

Barzen-Hanson, K.A., Roberts, S.C., Choyke, S., Oetjen, K., McAlees, A., Riddell, N., McCrindle, R., Ferguson, P.L., Higgins, C.P., Field, J.A., 2017. Discovery of 40 Classes of Per- and Polyfluoroalkyl Substances in Historical Aqueous Film-Forming Foams (AFFFs) and AFFF-Impacted Groundwater. Environ. Sci. Technol. 51, 2047–2057. https://doi.org/10.1021/acs.est.6b05843

Das, P., Arias, E., Kambala, V., Mallavarapu, M., Naidu, R., 2013. Remediation of perfluorooctane sulfonate in contaminated soils by modified clay adsorbent - A risk-based approach topical collection on remediation of site contamination. Water. Air. Soil Pollut. 224. https://doi.org/10.1007/s11270-013-1714-y

Filipovic, M., Woldegiorgis, A., Norström, K., Bibi, M., Lindberg, M., Österås, A.-H., 2015. Historical usage of aqueous film forming foam: A case study of the widespread distribution of perfluoroalkyl acids from a military airport to groundwater, lakes, soils and fish. Chemosphere 129, 39–45. https://doi.org/10.1016/j.chemosphere.2014.09.005

Gellrich, V., Brunn, H., Stahl, T., 2013. Perfluoroalkyl and polyfluoroalkyl substances (PFASs) in mineral water and tap water. J. Environ. Sci. Health - Part ToxicHazardous Subst. Environ. Eng. 48, 129–135. https://doi.org/10.1080/10934529.2013.719431

Gitipour, S., Farvash, E.S., Keramati, N., Yaghoobzadeh, P., Rezaee, M., 2015. Remediation of petroleum contaminated soils in urban areas using thermal desorption. J. Environ. Stud. 41, 643–652.

Gobelius, L., Hedlund, J., Dürig, W., Tröger, R., Lilja, K., Wiberg, K., Ahrens, L., 2018. Per- and Polyfluoroalkyl Substances in Swedish Groundwater and Surface Water: Implications for Environmental Quality Standards and Drinking Water Guidelines. Environ. Sci. Technol. https://doi.org/10.1021/acs.est.7b05718

Gobelius, L., Lewis, J., Ahrens, L., 2017. Plant Uptake of Per- and Polyfluoroalkyl Substances at a Contaminated Fire Training Facility to Evaluate the Phytoremediation Potential of Various Plant Species. Environ. Sci. Technol. 51, 12602–12610. https://doi.org/10.1021/acs.est.7b02926

Guelfo, J.L., Higgins, C.P., 2013. Subsurface Transport Potential of Perfluoroalkyl Acids at Aqueous Film-Forming Foam (AFFF)-Impacted Sites. Environ. Sci. Technol. 47, 4164–4171. https://doi.org/10.1021/es3048043

Guemiza, K., Coudert, L., Metahni, S., Mercier, G., Besner, S., Blais, J.-F., 2017. Treatment technologies used for the removal of As, Cr, Cu, PCP and/or PCDD/F from contaminated soil: A review. J. Hazard. Mater. 333, 194–214. https://doi.org/10.1016/j.jhazmat.2017.03.021

Hale, S.E., Arp, H.P.H., Slinde, G.A., Wade, E.J., Bjørseth, K., Breedveld, G.D., Straith, B.F., Moe, K.G., Jartun, M., Høisæter, Å., 2017. Sorbent amendment as a remediation strategy to reduce PFAS mobility and leaching in a contaminated sandy soil from a Norwegian firefighting training facility. Chemosphere 171, 9–18. https://doi.org/10.1016/j.chemosphere.2016.12.057

Heads of EPAs Australia and New Zealand (HEPA), 2018. PFAS National Environmental Management Plan.

Kupryianchyk, D., Hale, S.E., Breedveld, G.D., Cornelissen, G., 2016a. Treatment of sites contaminated with perfluorinated compounds using biochar amendment. Chemosphere, Biochars multifunctional role as a novel technology in the agricultural, environmental, and industrial sectors 142, 35–40. https://doi.org/10.1016/j.chemosphere.2015.04.085

Kupryianchyk, D., Hale, S.E., Breedveld, G.D., Cornelissen, G., 2016b. Treatment of sites contaminated with perfluorinated compounds using biochar amendment. Chemosphere, Biochars multifunctional role as a novel technology in the agricultural, environmental, and industrial sectors 142, 35–40. https://doi.org/10.1016/j.chemosphere.2015.04.085

Li, Y., Fletcher, T., Mucs, D., Scott, K., Lindh, C.H., Tallving, P., Jakobsson, K., 2018. Half-lives of PFOS, PFHxS and PFOA after end of exposure to contaminated drinking water. Occup. Environ. Med. 75, 46–51. https://doi.org/10.1136/oemed-2017-104651

Loganathan, B.G., Sajwan, K.S., Sinclair, E., Senthil, K., Kannan, K., 2007. Perfluoroalkyl sulfonates and perfluorocarboxylates in two wastewater treatment facilities in Kentucky and Georgia. Water Res. 41, 4611–4620. https://doi.org/10.1016/j.watres.2007.06.045

Mahinroosta, R., Senevirathna, L., 2020. A review of the emerging treatment technologies for PFAS contaminated soils. J. Environ. Manage. 255, 109896. https://doi.org/10.1016/j.jenvman.2019.109896

McCleaf, P., Englund, S., Östlund, A., Lindegren, K., Wiberg, K., Ahrens, L., 2017. Removal efficiency of multiple poly- and perfluoroalkyl substances (PFASs) in drinking water using granular activated carbon (GAC) and anion exchange (AE) column tests. Water Res. 120, 77–87. https://doi.org/10.1016/j.watres.2017.04.057

McGregor, R., 2018. In Situ treatment of PFAS-impacted groundwater using colloidal activated Carbon. Remediation 28, 33–41. https://doi.org/10.1002/rem.21558

Mejia-Avendaño, S., Munoz, G., Sauvé, S., Liu, J., 2017. Assessment of the Influence of Soil Characteristics and Hydrocarbon Fuel Cocontamination on the Solvent Extraction of Perfluoroalkyl and Polyfluoroalkyl Substances. Anal. Chem. 89, 2539–2546. https://doi.org/10.1021/acs.analchem.6b04746

Mulligan, C.N., Yong, R.N., Gibbs, B.F., 2001. Remediation technologies for metal-contaminated soils and groundwater: An evaluation. Eng. Geol. 60, 193–207. https://doi.org/10.1016/S0013-7952(00)00101-0

Murakami, M., Kuroda, K., Sato, N., Fukushi, T., Takizawa, S., Takada, H., 2009. Groundwater Pollution by Perfluorinated Surfactants in Tokyo. Environ. Sci. Technol. 43, 3480–3486. https://doi.org/10.1021/es803556w

Qi, Z., Chen, T., Bai, S., Yan, M., Lu, S., Buekens, A., Yan, J., Bulmǎu, C., Li, X., 2014. Effect of temperature and particle size on the thermal desorption of PCBs from contaminated soil. Environ. Sci. Pollut. Res. 21, 4697–4704. https://doi.org/10.1007/s11356-013-2392-4

Ross, I., McDonough, J., Miles, J., Storch, P., Thelakkat Kochunarayanan, P., Kalve, E., Hurst, J., S. Dasgupta, S., Burdick, J., 2018. A review of emerging technologies for remediation of PFASs. Remediat. J. 28, 101–126. https://doi.org/10.1002/rem.21553

Sörengård, M., Kleja, D.B., Ahrens, L., 2019a. Stabilization and solidification remediation of soil contaminated with poly- and perfluoroalkyl substances (PFASs). J. Hazard. Mater. 367, 639–646. https://doi.org/10.1016/j.jhazmat.2019.01.005

Sörengård, M., Niarchos, G., Jensen, P.E., Ahrens, L., 2019b. Electrodialytic per- and polyfluoroalkyl substances (PFASs) removal mechanism for contaminated soil. Chemosphere 232, 224–231. https://doi.org/10.1016/j.chemosphere.2019.05.088

Tröger, R., Klöckner, P., Ahrens, L., Wiberg, K., 2018. Micropollutants in drinking water from source to tap - Method development and application of a multiresidue screening method. Sci. Total Environ. 627, 1404–1432. https://doi.org/10.1016/j.scitotenv.2018.01.277

Vecitis, C.D., Park, H., Cheng, J., Mader, B.T., Hoffmann, M.R., 2009. Treatment technologies for aqueous perfluorooctanesulfonate (PFOS) and perfluorooctanoate (PFOA). Front. Environ. Sci. Eng. China 3, 129–151. https://doi.org/10.1007/s11783-009-0022-7

Wang, F., Lu, X., Li, X., Shih, K., 2015. Effectiveness and Mechanisms of Defluorination of Perfluorinated Alkyl Substances by Calcium Compounds during Waste Thermal Treatment. Environ. Sci. Technol. 49, 5672–5680. https://doi.org/10.1021/es506234b

Wang, F., Shih, K., Lu, X., Liu, C., 2013. Mineralization Behavior of Fluorine in Perfluorooctanesulfonate (PFOS) during Thermal Treatment of Lime-Conditioned Sludge. Environ. Sci. Technol. 47, 2621–2627. https://doi.org/10.1021/es305352p

Watanabe, N., Takemine, S., Yamamoto, K., Haga, Y., Takata, M., 2016. Residual organic fluorinated compounds from thermal treatment of PFOA, PFHxA and PFOS adsorbed onto granular activated carbon (GAC). J. Mater. Cycles Waste Manag. 18, 625–630. https://doi.org/10.1007/s10163-016-0532-x

Yamada, T., Taylor, P.H., Buck, R.C., Kaiser, M.A., Giraud, R.J., 2005. Thermal degradation of fluorotelomer treated articles and related materials. Chemosphere 61, 974–984. https://doi.org/10.1016/j.chemosphere.2005.03.025

Yao, Y., Volchek, K., Brown, C.E., 2015. Characterization of environmental samples from an aqueous film forming foam (AFFF)-impacted firefighting training area in Canada. Presented at the Proceedings of the 38th AMOP Technical Seminar on Environmental Contamination and Response, pp. 265–282.

---

## [Decision Letter · Decision Letter 1]

12 May 2020

PONE-D-19-33947R1

Thermal desorption as a high removal remediation technique for soils contaminated with per- and polyfluoroalkyl substances (PFASs)

PLOS ONE

Dear Mr sörengård,

Thank you for submitting your manuscript to PLOS ONE. After careful consideration, we feel that it has merit but does not fully meet PLOS ONE’s publication criteria as it currently stands. Therefore, we invite you to submit a revised version of the manuscript that addresses the points raised during the review process.

Please note the revisions recommended by the reviewer and please note the additional clarifications regarding why these revisions are required by this reviewer.

We would appreciate receiving your revised manuscript by Jun 26 2020 11:59PM. To enhance the reproducibility of your results, we recommend that if applicable you deposit your laboratory protocols in protocols.io, where a protocol can be assigned its own identifier (DOI) such that it can be cited independently in the future. For instructions see: http://journals.plos.org/plosone/s/submission-guidelines#loc-laboratory-protocols

We look forward to receiving your revised manuscript.

Kind regards,

Jamie C. DeWitt

Academic Editor

PLOS ONE

Additional Editor Comments (if provided):

Please accept our apologies for the length of the review of your manuscript. It was challenging to find reviewers and now due to COVID, schedules are even more challenging. A few additional revisions have been requested.

Reviewers' comments:

Reviewer's Responses to Questions

**Comments to the Author**

1. If the authors have adequately addressed your comments raised in a previous round of review and you feel that this manuscript is now acceptable for publication, you may indicate that here to bypass the “Comments to the Author” section, enter your conflict of interest statement in the “Confidential to Editor” section, and submit your "Accept" recommendation.

Reviewer #1: (No Response)

Reviewer #2: All comments have been addressed

2. Is the manuscript technically sound, and do the data support the conclusions?

Reviewer #1: Partly

Reviewer #2: Yes

3. Has the statistical analysis been performed appropriately and rigorously? 

Reviewer #1: Yes

Reviewer #2: Yes

4. Have the authors made all data underlying the findings in their manuscript fully available?

Reviewer #1: Yes

Reviewer #2: Yes

5. Is the manuscript presented in an intelligible fashion and written in standard English?

Reviewer #1: Yes

Reviewer #2: Yes

6. Review Comments to the Author

Reviewer #1: General

I would like to extend my thanks to the authors for their sincere effort to address the comments provided regarding the original version of the manuscript. I can see that the comments were well-received and that there was an attempt to address each one thoroughly. The result is an overall improvement; however, there are instances where I am not sure the intent of my comment was fully understood and/or where the resulting edits are somewhat confusing. To hopefully help address this I have tried to clarify some of these points in the line by line comments below.

Title/Abstract

• Line 19: Here and throughout the document, I recognize that the authors changed use of the word efficiency to rate in response to a prior comment, but I do not agree with use of the word rate. To me, removal rate implies a parameter with units that include time, and in this work it is being used to refer to the fraction of PFAS desorbed. I suggest changing to “fraction removed” or similar.

• Line 20: The authors refer to “naturally contaminated soil.” Here and throughout the paper, I suggest that they modify this to field contaminated or similar. Naturally applies that the source of the impact is not anthropogenic in origin, and all PFAS are man-made.

Introduction

• Lines 37-40: “The sentence beginning Thus, there is an urgent need to remediate….,” needs to be reworded. I am not certain if I understand the intent of the sentence, but I think it could be fixed by rewording it to say, “…PFAS-contaminated hotspot areas despite the fact that few soil guideline values…”

• Line 42: I recommend changing “problem owners” here and throughout the manuscript to “responsible parties.”

Results and Discussion

• Lines 141-142 state that removal rate generally increases with increasing temperature. In the PFAS fortified clay soil, the PFAS concentration at 150 has increased relative to the reference concentrations. Given that this soil was laboratory fortified, I assume that unknown precursors were not present in this sample that could lead to this increase. Were there background concentrations in the clay that were not originally accounted for?

• Lines 163-165: What about the potential effects of organic carbon?

• Line 170: I believe the authors mean to say “With the exception of PFBA…”

• Lines 172-173: “This indicates that the main removal mechanism observed in this study is thermal desorption and not degradation.“ The lack of increase in shorter chain homologues is not sufficient evidence to support desorption over degradation. For example, what if the process generated short-chain PFAS smaller than those monitored in this project?

• Lines 177-178 state that the optimal treatment temperature is 350-450oC; however line 182 says that PFAS-impacted soils can be treated using “thermal desorption at temperatures above 450oC. This is contradictory.

• Lines181-182: In the previous version of the manuscript, I asked that the authors define their criteria for efficient treatment. This had the unintended consequence of leading the authors to change all uses of “efficient” in the manuscript. As mentioned above, I do not think the use of the word rate herein is the best way to express what the authors actually evaluated which is the fraction removed and not, for example, first order rate coefficients. In this specific sentence the authors now say soils can be treated with “high removal,” which doesn’t address the issue. What are their criteria defining high? I recommend the authors stick to the facts of their data. For example, they might say that this treatment technique is capable of removing up to x% of PFAS in soils.

Reviewer #2: (No Response)

7. PLOS authors have the option to publish the peer review history of their article (what does this mean?). If published, this will include your full peer review and any attached files.

Reviewer #1: No

Reviewer #2: No

---

## [Author Response · Author response to Decision Letter 1]

13 May 2020

Response to reviewers

Reviewer #1: General

I would like to extend my thanks to the authors for their sincere effort to address the comments provided regarding the original version of the manuscript. I can see that the comments were well-received and that there was an attempt to address each one thoroughly. The result is an overall improvement; however, there are instances where I am not sure the intent of my comment was fully understood and/or where the resulting edits are somewhat confusing. To hopefully help address this I have tried to clarify some of these points in the line by line comments below.

We thank the reviewers for the thorough comments to improve the manuscript

Title/Abstract

• Line 19: Here and throughout the document, I recognize that the authors changed use of the word efficiency to rate in response to a prior comment, but I do not agree with use of the word rate. To me, removal rate implies a parameter with units that include time, and in this work it is being used to refer to the fraction of PFAS desorbed. I suggest changing to “fraction removed” or similar.

Adressed

• Line 20: The authors refer to “naturally contaminated soil.” Here and throughout the paper, I suggest that they modify this to field contaminated or similar. Naturally applies that the source of the impact is not anthropogenic in origin, and all PFAS are man-made.

Adressed

Introduction

• Lines 37-40: “The sentence beginning Thus, there is an urgent need to remediate….,” needs to be reworded. I am not certain if I understand the intent of the sentence, but I think it could be fixed by rewording it to say, “…PFAS-contaminated hotspot areas despite the fact that few soil guideline values…”

Adressed

• Line 42: I recommend changing “problem owners” here and throughout the manuscript to “responsible parties.”

Adressed

Results and Discussion

• Lines 141-142 state that removal rate generally increases with increasing temperature. In the PFAS fortified clay soil, the PFAS concentration at 150 has increased relative to the reference concentrations. Given that this soil was laboratory fortified, I assume that unknown precursors were not present in this sample that could lead to this increase. Were there background concentrations in the clay that were not originally accounted for?

The background concentration was accounted for, and would have been shown in the untreated reference soil. The difference is within the measurement errors of the individual PFASs.

• Lines 163-165: What about the potential effects of organic carbon?

It is well known that PFAS partition to a higher extent to soil organic carbon, and the variable could have significant effect on the thermal desorption process. However, this effect was not isolated in this experiment and from the three different soils the effect could not be elucidated, and hence outside the scope of this study.

• Line 170: I believe the authors mean to say “With the exception of PFBA…”

Adressed

• Lines 172-173: “This indicates that the main removal mechanism observed in this study is thermal desorption and not degradation.“ The lack of increase in shorter chain homologues is not sufficient evidence to support desorption over degradation. For example, what if the process generated short-chain PFAS smaller than those monitored in this project?

From the previous studies by our research group 1, and other groups 2, degradation occurs by stepwise defluorination and increase of some of shorter chained PFASs included in the study is normally observed (i.e. PFBA and PFPeA) degradation products from longer chained PFASs. We can not say for sure that no ultra-short chained PFASs have been formed, but there was no increase in i.e. PFBA or PFPeA or other PFASs and hence no indications of PFAS degradation. Hence, we opt to keep the sentence as it is.

• Lines 177-178 state that the optimal treatment temperature is 350-450oC; however line 182 says that PFAS-impacted soils can be treated using “thermal desorption at temperatures above 450oC. This is contradictory.

Adressed

• Lines181-182: In the previous version of the manuscript, I asked that the authors define their criteria for efficient treatment. This had the unintended consequence of leading the authors to change all uses of “efficient” in the manuscript. As mentioned above, I do not think the use of the word rate herein is the best way to express what the authors actually evaluated which is the fraction removed and not, for example, first order rate coefficients. In this specific sentence the authors now say soils can be treated with “high removal,” which doesn’t address the issue. What are their criteria defining high? I recommend the authors stick to the facts of their data. For example, they might say that this treatment technique is capable of removing up to x% of PFAS in soils.

The authors agree with the reviewer that “removal rates” and “removal efficiency” can be misleading concepts, and have subsequently replaced the term with either “removal” or “fraction removed”.

However, in comparison to multiple other studies and the long experience of the authors of working with PFAS remediation in both water and soil matrices, the authors consider removal rates of >99% to be relatively high removal compared to other techniques. This is an important finding of the study, and hence the authors keep the “high removal” expression, while also throughout the manuscript the absolute removal is clearly expressed in %.

1. Franke, V., Schäfers, M. D., Lindberg, J. J. & Ahrens, L. Removal of per- And polyfluoroalkyl substances (PFASs) from tap water using heterogeneously catalyzed ozonation. Environ. Sci. Water Res. Technol. 5, 1887–1896 (2019).

2. Liu, Y. et al. Efficient Mineralization of Perfluorooctanoate by Electro-Fenton with H2O2 Electro-generated on Hierarchically Porous Carbon. Environ. Sci. Technol. 49, 13528–13533 (2015).

---

## [Editor Report · Decision Letter 2]

28 May 2020

Thermal desorption as a high removal remediation technique for soils contaminated with per- and polyfluoroalkyl substances (PFASs)

PONE-D-19-33947R2

Dear Dr. sörengård,

We are pleased to inform you that your manuscript has been judged scientifically suitable for publication and will be formally accepted for publication once it complies with all outstanding technical requirements.

With kind regards,

Jamie C. DeWitt

Academic Editor

PLOS ONE

Additional Editor Comments (optional):

Thank you for addressing reviewer concerns and for being patient with the timeline.
---

## [Editor Report · Acceptance letter]

3 Jun 2020

PONE-D-19-33947R2 

Thermal desorption as a high removal remediation technique for soils contaminated with per- and polyfluoroalkyl substances (PFASs) 

Dear Dr. Sörengård:

I'm pleased to inform you that your manuscript has been deemed suitable for publication in PLOS ONE. Congratulations! Your manuscript is now with our production department. 

Kind regards, 

on behalf of

Dr. Jamie C. DeWitt 

Academic Editor

PLOS ONE